# PHB Producing Cyanobacteria Found in the Neighborhood—Their Isolation, Purification and Performance Testing

**DOI:** 10.3390/bioengineering9040178

**Published:** 2022-04-18

**Authors:** Katharina Meixner, Christina Daffert, Lisa Bauer, Bernhard Drosg, Ines Fritz

**Affiliations:** 1Department of Agrobiotechnology IFA-Tulln, Institute of Environmental Biotechnology, University of Natural Resources and Life Sciences, Vienna, Konrad-Lorenz-Straße 20, 3430 Tulln, Austria; katharina.meixner@boku.ac.at (K.M.); christina.daffert@boku.ac.at (C.D.); bernhard.drosg@boku.ac.at (B.D.); 2BEST Bioenergy and Sustainable Technologies GmbH, Inffeldgasse 21b, 8010 Graz, Austria; lisa.bauer@best-research.eu

**Keywords:** cyanobacteria, habitat conditions, sampling, wild types, single species selection, purification, axenic cultures, growth, PHB

## Abstract

Cyanobacteria are a large group of prokaryotic microalgae that are able to grow photo-autotrophically by utilizing sunlight and by assimilating carbon dioxide to build new biomass. One of the most interesting among many cyanobacteria cell components is the storage biopolymer polyhydroxybutyrate (PHB), a member of the group of polyhydroxyalkanoates (PHA). Cyanobacteria occur in almost all habitats, ranging from freshwater to saltwater, freely drifting or adhered to solid surfaces or growing in the porewater of soil, they appear in meltwater of glaciers as well as in hot springs and can handle even high salinities and nutrient imbalances. The broad range of habitat conditions makes them interesting for biotechnological production in facilities located in such climate zones with the expectation of using the best adapted organisms in low-tech bioreactors instead of using “universal” strains, which require high technical effort to adapt the production conditions to the organism‘s need. These were the prerequisites for why and how we searched for locally adapted cyanobacteria in different habitats. Our manuscript provides insight to the sites we sampled, how we isolated and enriched, identified (morphology, 16S rDNA), tested (growth, PHB accumulation) and purified (physical and biochemical purification methods) promising PHB-producing cyanobacteria that can be used as robust production strains. Finally, we provide a guideline about how we managed to find potential production strains and prepared others for basic metabolism studies.

## 1. Introduction

The worldwide growing plastic pollution has initiated a search for plastic-like materials which are biodegradable in organic recovery facilities (well managed industrial compost) as well as in the environment. Among others, bacteria storage polymers of the type polyhydroxyalkanoate (PHA) are promising candidates [1,2]. Biotechnological production processes have already been developed that can utilize industrial waste processes (instead of agricultural raw material) as resources [3], and many of those processes have been optimized for improved productivity [4]. A production with photo-autotrophic bacteria, such as cyanobacteria, will even avoid the dependency on waste and residues, because carbon dioxide (CO_2_) can be utilized as sole carbon source [5]. Cyanobacteria are known to produce the homo-polymer of hydroxybutyrate, PHB, which is only one of 150 different types of PHAs identified so far [6,7,8].

Well-investigated strains of cyanobacteria can be obtained from culture collections, such as DSMZ, PCC, ATCC, CCALA to name only a few. The benefit of using such strains is in the public availability of comprehensively documented properties and cellular compositions, which makes them something like a standard—further research work does not need to start from scratch. As lucrative as those strains are for basic science studies, they are rarely suitable for biotechnological use of a bigger scale, when ambient conditions, like the influence of climate or the presence of endemic competitors and predators, cannot be controlled—at least not with justified effort.

Microorganisms that grow in significant abundance in a certain natural habitat are, most likely, well adapted to the local conditions and have developed properties that makes them superior compared to the other members of the biocenosis [9]. Temperature and average light energy emission are the most important factors in prospect of a future use in biotechnology; the ideal composition of minerals and trace elements can be adjusted in the cultivation medium.

When intending to produce PHB with photo-autotrophically grown cyanobacteria [10], production effort and production cost will decide if the process can be operated in an economical way to compete with established market prices [11,12,13]. Circulation of production media, mostly water and mineral nutrients will decrease the production cost [14]. Harvest and utilization of other valuable cyanobacteria cell components, such as vitamins, unsaturated fatty acids or pigments [15] can help to make a phototrophic production economically viable [16].

Temperature requirements far off from ambient conditions and the operation of closed and sterile bioreactors will increase the production effort even more and are, most likely, only viable for producing special chemicals for use in pharmaceutical or cosmetic applications. Any cyanobacteria strain that can grow at local ambient conditions as single species but in non-axenic cultivation will help to decrease costs and is worth testing and characterizing for its key properties.

Pros and cons about using wild strains:(+) Adapted to local climate, especially considering daily and annual temperature range and sunlight availability,(+) When found as abundant species in a local habitat, it will have a growth advantage over local competitors,(+) Resistance against local predators and pathogens,(+) Can be operated non-axenically (in open bioreactors),(+) No ecological harm from spoilage after leaks or accidents—no special admission procedure,(+) Free of third-party rights,(−) Only small chance of superior strain productivity,(−) High effort for strain characterization and improvement, if necessary.

Understanding the metabolic pathways and the relation between PHB synthesis and other parts of the anabolism will allow improving cyanobacterial production processes without genetic manipulation [17]. Despite increasing knowledge, expressed in a high number of publications about PHB producing cyanobacteria grown in shaking flasks in laboratory environment, comparably little is published about scale-up [18]. Growing cyanobacteria in bigger volumes in open photobioreactors under non-sterile and, probably, under not so well controlled conditions often fails and rarely succeeds [10,19].

In contrast to production goals, axenic cultures are needed for applications where the coexistence of other organisms is undesirable. Such include genome sequencing, identifying producers of bioactive compounds, or building an artificial consortium for bioremediation, as well as clarifying the relationships between microalgae and other organisms using -omics tools [20,21]. A prerequisite for applied research is knowledge about the origin or about the cause of an observed effect and, finally, to differentiate between behaviors of cyanobacteria and those of an uncontrolled heterotroph accompanying flora.

Axenic culture establishment is the process of eliminating undesirable organisms (contaminants) to obtain a viable culture of desired organisms (quarry). The strategy depends strongly on the characteristics of the quarry and on the contaminants in each microbial community [20].

The aim of this work was to isolate cyanobacteria strains that accumulate PHB and are robust enough for cultivation at outdoor conditions in an low-tech open reactor. Axenic cultures should be generated from the most promising of those isolates for subsequent characterization of their growth kinetics and for molecular analysis.

For this purpose, samples were taken at 27 locations around the world representing various climatic regions. Cyanobacteria strains were isolated, identified, screened and evaluated for growth and for PHB accumulation. Finally, we applied different methods to find the best suitable strategy to obtain axenic cultures. Therefore, physical and chemical separating and purifying methods were carried out.

## 2. Materials and Methods

### 2.1. Sampling in the Wild and Strains from Culture Collections

All samples were generally treated in the same way. Liquid samples were transferred to mineral medium [22] on site. In case of a short travel distance, the samples were brought to the lab immediately, in other cases samples were stored at windows in hotel rooms until transport to the lab. The cultures were incubated in Erlenmeyer flasks at 21 °C, in elevated CO_2_ concentration (approx. 1.5 vol%), at 85–95 rpm (orbital shaker VKS 75 control, Edmund Bühler GmbH, Bodelshausen, Germany) and a day–night cycle of 16:8 h with an illumination of approx. 108 μmol photons m^−2^ s^−1^ (metal halide lamp, Philips Master HPI-T Plus, 250 W) until the first green coloration visible to the bare eye. After that, cultures were inoculated into fresh BG-11 medium approx. every three to six days to pre-select the fastest growing photo-autotrophs.

In addition to the isolated wild type strains, screening and methods for obtaining axenic cultures were also carried out with the strains *Synechocystis* sp. PCC6803 and *Synechocystis* cf. *salina* CCALA192 as reference, which were obtained from the Pasteur culture collection (FRA) and from the culture collection of autotrophic organisms (CZE), respectively, both in non-axenic form.

### 2.2. Growing Media

For different purposes described herein, different media were used. For selecting fast growing species, for separating into single-species cultures and for obtaining axenic cultures, a variation of BG-11 medium [22] was used, containing half of the recommended amount of sodium nitrate (code: BG-I) [23]. For PHB screening experiments as well as for evaluating growth and PHB, a nutrient limited mineral medium, also based on BG-11, was used. This nutrient limited medium (code: 22O) contained following ingredients per liter: NaNO_3_: 0.45 g, Fe(NO_3_)·39H_2_O: 0.025 g, MgSO_4_·7H_2_O: 0.10 g, CaCl_2_·2H_2_O: 0.60 g, Na_2_CO_3_: 0.20 g, K_2_HPO_4_: 0.08 g, trace element solution 1.50 mL. Composition of the trace element solution per liter: H_3_BO_3_: 0.509 g, CuSO_4_·5H_2_O: 0.150 g, KI: 0.181 g, FeCl_3_·6H_2_O: 0.293 g, MnSO_4_·H_2_O: 0.296 g, Na_2_MoO_4_·2H_2_O: 0.082 g, NiSO_4_·6H_2_O: 0.275 g, Co(NO_3_)_2_·6H_2_O: 0.100 g, ZnSO_4_·7H_2_O: 0.490 g, KAl(SO_4_)_2_·12H_2_O: 0.395 g, KCr(SO_4_)_2_·12H_2_O: 0.470 g [23].

Besides the liquid media, agar-plates with BG-I medium were used. For fractionated streaks, plates with BG-I containing 1.5% agar were prepared, for phototaxis experiments 0.4% or 1% agar was added to BG-I.

Depending on the approach, different active substances were added to the liquid or solid cultivation medium. Cycloheximide, which inhibits cytoplasmic protein synthesis in eukaryotes [24], was used to suppress eukaryotic microalgae in the culture. To obtain axenic cultures lysozyme, imipenem, penicillin G and streptomycin were used. Lysozyme lyses peptidoglycan of the cell wall of bacteria [25]. Imipenem, a broad-spectrum β-lactam antibiotic, inhibits the biosynthesis of bacterial peptidoglycan [26]. Penicillin G too is a β-lactam antibiotic, it inhibits the formation of the bacterial cell wall and acts on G+ bacterial. Streptomycin is an aminoglycoside antibiotic that inhibits protein synthesis in prokaryotic ribosome [27]. Besides that, glucose and LB medium were added to liquid and solid media in the antibiotic approaches (BG-I+ medium).

### 2.3. Selecting Fast Growing Species

Samples were cultivated in a mineral medium without carbon source (see Section 2.2) to get rid of the majority of heterotrophic bacteria and multicellular organisms. The cultures were transferred in short intervals (as soon as cell suspensions turned light green) into fresh media to select the fastest growing strains. Such media transfers were repeated as often as necessary, up to 12 times.

### 2.4. Separation into Single-Species Cultures and Identification

Once decreased numbers of morphologically different cyanobacteria were visible under the microscope, additional selection criteria were applied: (i) mixed incubation in BG-I medium with cycloheximide (100 mg L^−1^) to suppress growth of eukaryotic microorganisms and (ii) plating on BG-I agar medium (fractionated streaks) with colonies re-inoculated in new liquid BG-I medium.

Morphological identification of non-axenic single-species strains and purity controls of the samples were carried out at bright field and phase contrast microscopy at 40-fold magnification (Olympus BX43, Austria, Vienna). Pictures were taken with a digital camera (Canon EOS 1300D).

For identification via 16S-rDNA, fragments of approx. 700 bp were used as obtained from PCR amplification using the primers 5′-CGGACGGGTGAGTAACGCGTGA-3′ (CYA 106F) and 5′-GACTACTGGGGTATCTAATCCCATT-3′ (CYA 781R(a)), which are specific for cyanobacteria [28,29]. The protocol of Nübel et al. [29] was followed with a higher reaction volume to gain a sufficient amount of DNA for sequencing [28].

DNA sequencing was carried out by GATC Biotech AG (European Genome and Diagnostics Center, Constance, Germany) and LGC Genomics (http://www.lgcgroup.com/; accessed on 24 February 2022). Subsequently, DNA sequences were compared via NCBI BLAST.

### 2.5. Evaluation of Growth and PHB Accumulation

Cyanobacteria strains were grown in nutrient limited mineral medium 22O, which is based on BG-I, but with reduced amounts of nitrogen and phosphorous to reach starvation at roughly 1 g dry biomass/liter. Reaching nutrient starvation, the cells started to change color from bluish-green over olive-green to orange. For a fast screening, Nile-red staining and fluorescence microscopy (Olympus AH-3 AHBT3 VANOX, RFL-T3 Transformer, Hg-burner, Leica EC3 camera) were carried out using the culture suspensions without further pretreatment.

PHB producing strains were investigated in more detail by recording growth curves and PHB accumulation kinetics. The cultures were grown about 21 days and were monitored via periodical measurement of optical density (OD), cell dry weight (CDW) and PHB content.

#### 2.5.1. Nile-Red Staining

For PHB staining, Nile-red solution (50 µg mL^−1^ in 98% ethanol) was directly added to the cell suspension. After 10 min reaction time, 20 µL mixture were distributed on a glass slide and heat fixed for fluorescence microscopy with 450–500 nm excitation and >550 nm emission wavelengths selected.

#### 2.5.2. Growth Analysis

For evaluating the growth optical density (OD) was measured at 435 nm, 485 nm, 680 nm and 750 nm with an UV-VIS Spectrophotometer (Shimadzu UV-1800, Kyoto, Japan). Furthermore, the cellular dry weight (CDW) was measured from 10 mL cell suspension, which was centrifuged, washed and dried at 105 °C.

#### 2.5.3. PHB Analysis

The PHB content of the biomass was analyzed based on the method described in [30]. Therefore, 5 to 10 mL cell suspension were washed and dried (105 °C) and subsequently digested with concentrated (98%) sulfuric acid (H_2_SO_4_) for 30 min at 90 °C. Afterwards, the samples were filled to 10 mL with deionized water and prepared for HPLC (high-performance liquid chromatography) analysis (Agilent 1100; column: Transgenomic COREGEL 87H3; detector: Agilent 1100 RI, Santa Clara, CA, USA).

### 2.6. Obtaining Axenic Cultures

Multiple approaches were tried to reduce heterotrophic accompanying flora or, in the best case, to remove it totally in order to obtain an axenic cyanobacteria culture. Those approaches were: separating methods (agar-based methods, phototaxis, micro-pipetting via cell sorter and micromanipulator), serial dilutions (MPN approach), as well as killing methods (addition of cycloheximide, lysozyme, antibiotics, incubation at elevated temperature and salinity).

#### 2.6.1. Phototaxis

For this approach, which was tested with PCC6803, CCALA192 and *Synechocystis* sp. IFA-3 two kinds of agar-plates were prepared, one with 0.4% and one with 1% final agar concentration. On these agar plates samples (10 µL cyanobacteria culture) and glucose (5 µL, 1% solution) were applied in two small holes stuck with a glass Pasteur pipette (Figure 1). The plates were then wrapped into aluminum foil and a small hole was made (opposite the glucose spot) so that a small light beam could reach the plate. The plates were incubated for 6 days at 22 °C and light (day–night cycle: 16:8 h, light intensity: approx. 108 μmol photons m^−2^ s^−1^, metal halide lamp, Philips Master HPI-T Plus, 250 W). The experiments were carried out in triplicates.

#### 2.6.2. Antibiotic Treatment

This approach was carried out with the *Synechocystis* strains CCALA192, *Synechocystis* sp. IFA-3 and PCC6803. The methods changed in response to first results. For the approach with PCC6803, 1 g lysozyme, 100 mg imipenem and 100 mg streptomycin per liter were added to 10 mL cell suspension, which was then incubated 4 h in the dark [31]. Subsequently, cells were washed and resuspended in mineral medium to which glucose (1 g 100 mL^−1^) and LB-medium (10 g 100 mL^−1^) were added before it was again incubated in the dark at 22 °C for 30 min [27]. Then, again antibiotics were added (100 mg imipenem and 100 mg streptomycin per liter) and incubated in the dark for 18 h at 22 °C.

For the approach with CCALA192 and *Synechocystis* sp. IFA-3 10 mL cell suspension were incubated for 12 to 24 h in the dark, then the cell number was counted (counting chamber Neubauer improved, Marienfeld Germany, depth: 0.01 mm, area: 0.0025 mm^2^). Afterwards, an organic carbon source (1.2 g glucose per 100 mL cell suspension) was added [32] and to the reference the same amount on RO water. The cultures were incubated in the dark at 22 °C and approx. 85 rpm (standard orbital shaker Model 1000, VWR^®^) over night. Afterwards, 100 mg imipenem per liter and 100 mg penicillin G were added [32]. Again, the cultures were incubated in the dark at 22 °C and 85–95 rpm (orbital shaker VKS 75 control, Edmund Bühler GmbH) over night.

All cultures were finally plated out the next day. Therefore, the cells were washed and re-suspended in BG-I medium before plating 50 µL on agar. The agar plates were based on BG-I and BG-I + 0.5% glucose medium, respectively. Besides that, 0.5 mL were transferred into 100 mL flasks containing 30 mL BG-I and BG-I + 0.5% glucose medium, respectively. Plates and flasks were incubated at 22 °C and a day–night cycle of 16:8 h with an illumination of approx. 108 μmol photons m^−2^ s^−1^ (metal halide lamp, Philips Master HPI-T Plus, 250 W, Philips, Amsterdam, The Netherlands) and controlled after 2 and 20 h and then 2 to 3 times per week until growth was visible. In each run, a blank was included, instead of antibiotics RO-water was added. Antibiotics were sterile filtered and glucose solutions autoclaved separately. Approaches were carried out in triplicates.

#### 2.6.3. Elevated Temperature and Salinity

Cultures of PCC6803 as well as isolated cyanobacterial accompanying flora were inoculated in BG-I medium an in BG-I + 5 g L^−1^ glucose. The shaking flasks were put on a shaker (standard orbital shaker Model 1000, VWR^®^, Radnor, PA, USA) in a climate room at 37 °C and a day–night cycle of 16:8 h (75.9 ± 9.5 μmol photons m^−2^ s^−1^, LED warm white 3000 K, 12 VDC) for 13 days. The experiments were carried out in triplicates.

An additional approach was carried out with *Synechocystis* sp. IFA-3. The aim was to evaluate how *Synechocystis* sp. and its accompanying flora can cope with salt stress and elevated temperature. For this purpose, *Synechocystis* sp. (1.2 × 10^7^ cells) was inoculated in BG-I medium and BG-I medium with 4% (684 mM) NaCl. The 100-mL Erlenmeyer flasks containing 40 mL cell suspension were incubated on a shaker (standard orbital shaker Model 1000, VWR^®^) in a climate room at 37 °C and a day–night cycle of 16:8 h (75.9 ± 9.5 μmol photons m^−2^ s^−1^, LED warm white 3000 K, 12 VDC) for 13 days. The experiments were carried out in triplicates.

#### 2.6.4. Cell Sorter

Cell sorting experiments were carried out with single cellular, non-filamentous cyanobacteria using a Beckmann Coulter (LSR-System ASTRIOS, AU05037). The size of the cells in the samples was measured and those in the range of cyanobacterial cells (3 to 5 µm) were separated into small tubes and subsequently cultivated in BG-I medium. This approach was tested with the *Synechocystis* strains PCC6803, CCALA192 and IFA-3.

#### 2.6.5. Micromanipulator

The micromanipulator consisted of a holder for a homemade glass capillary with a small disposable syringe for minimal fluid movement. The glass capillaries were produced from Pasteur pipets to accommodate about 1–5 µL of fluid, according to anticipated needs. Areas of predominantly cyanobacterial cells were sought under the inverted microscope (Nikon TMS, Nikon, Tokyo, Japan), and in each case between one and up to 20 cells were aspirated and transferred to fresh medium.

#### 2.6.6. MPN Dilution Approach

Based on the count of cyanobacterial cells (counting chamber Neubauer improved, Marienfeld Germany, depth: 0.01 mm, area: 0.0025 mm^2^), dilutions were prepared to achieve a suspension of 6 cells in 2.4 mL (=2.5 mL^−1^). In each well of a 24-well plate, 2 mL BG-I medium were put and 0.1 mL of the final dilution was added with the prospect of having distributed 6 cells over 24 wells of the plate. The plate was incubated on a shaker (New Brunswick Scientific Co G25 Controlled Environment Incubator Shaker, approx. 85 rpm; standard orbital shaker Model 1000, VWR^®^, approx. 85 rpm) at 22 to 25 °C and 35 to 37 °C. After about 14 to 21 days, some wells turned slightly green. After about 28 days, cultures were microscopically controlled and putative axenic cultures were transferred into shaking flasks with BG-I medium and cultivated under the above described conditions. These approaches were carried out with the strains PCC6803, CCALA192 and *Synechocystis* sp. IFA-3.

## 3. Results and Discussion

### 3.1. Sampling

Sampling took place in different climatic regions, reaching from cold semi-arid climate (BSk), over warm-summer humid continental climate (Dfb) to tundra climate (ET) (examples in Figure 2); the classification is based on Köppen [33]. Samples from puddles, lakes, springs and rivers were directly scooped into a sterile plastic tube. When the samples could not be transported to the laboratory the same day, they were diluted with BG-I medium in equal volumes. Samples from solid surfaces were scratched off using a metal spatula and the harvested solids were suspended in sterile BG-I medium in a plastic tube on-site.

A total of 71 samples were taken from 25 locations (details in Table 1).

### 3.2. Selecting Fast Growing Species

After transport to the lab, the samples were transferred to shaking flasks with BG-I medium and cultivated under photo-autotrophic conditions at 21 °C 16 h light/8 h dark on an orbital shaker at ca. 85–90 rpm. As soon as a culture developed a green color, a small volume was transferred into another flask with BG-I medium. Culture transfers were repeated up to 12 times or until the culture was dominated by one or two morphologically different strains (microscopic control in advance of each transfer)—or until the cultures lost all of their green color.

In total, 34 growing mixed cultures (containing eukaryotic algae and cyanobacteria) were obtained (selected examples are shown in Figure 3).

### 3.3. Getting Rid of Eukaryotic Microalgae

Enriched fast growing phototrophs were transferred to a BG-I medium, which contained 100 mg L^−1^ cycloheximide, and were cultivated under the same conditions as before for 5–6 days. A preliminary experiment (data in Appendix A) demonstrated the effectiveness of the antibiotics. All wild type cultures that contained vital cyanobacteria after several transfers were treated that way. Eukaryotic algae free cultures were obtained in most cases (example in Figure 3B), but in some of the cultures we lost all cyanobacteria as well.

### 3.4. Separation into Single-Species Cultures

Not all cyanobacteria grew well on BG-I agar, only in some cases colony-like clusters of cyanobacteria could be obtained from fractionated streaks. However, multiple differently looking colonies were obtained from some samples and were harvested separately. All of those clusters contained cyanobacteria often mixed with an even bigger amount of heterotrophic colorless bacteria. Colonies were in most cases colored blue-green, as expected, but also olive-green and red-orange colonies were obtained (see examples in Figure 4). Samples from colonies were inspected in the microscope before transfer to new shaking flasks with BG-I medium. All such isolates were numbered in ascending order per sampling site.

Advantages of agar-based methods are that simple and macroscopic results are received and that axenic cultures can often be directly established without further treatment. We observed quite often long lag times until visible colonies were grown, in some cases up to four weeks. As mentioned before, no colonies were formed from some of the enriched cultures. In case the streak separates the cyanobacteria cells from essential synergists, the cyanobacteria may not grow or may even die out before they come into contact again [20].

### 3.5. Identification of Single-Species Cultures

All isolates and cultures had been routinely inspected microscopically, using bright field and phase contrast techniques (see Figure 3). Cyanobacteria had been characterized during all enrichment and isolation steps based on morphological characteristics. Although not axenic, DNA was extracted and cyanobacteria specific 16S rDNA [29] was amplified for sequencing. Most probable classifications and identifications are provided in Table 1, sequences are provided in Appendix C. We are aware that some of the sequences had less than 99% accordance. Therefore, we chose the most probable genera and/or species name combined from morphology and sequence accordance.

### 3.6. Screening of PHB Production

Single-species cyanobacteria cultures that grew in BG-I medium in shaking flasks to an optical density of at least 5 (ʎ = 435 nm) within 3 weeks were investigated for their ability to produce and accumulate PHB. For this purpose, inocula of the selected strains were transferred into the self-limiting mineral medium 22O and incubated at the same conditions as before. Cultures grown for four weeks were stained with Nile-red and microscopically inspected at fluorescence mode. Cultures containing at least some stained granules (see examples in Figure 5) were selected for quantitative PHB analysis and those which grew comparably fast (OD_435_ higher than 10 within two weeks) were also used for growth and PHB production experiments (results are shown in Table 1).

Passing all these previous steps, we obtained four potential production strains, one from the warm Mediterranean, two from cold arid and one from temperate oceanic climate zones. Unfortunately, we were not able to isolate thermophilic or psychrophilic cyanobacteria.

### 3.7. Evaluation of Growth and PHB Accumulation

The four strains *Chlorogloeopsis* sp. (isolate 4 from location Heinrichs, AUT), *Crocosphaera* sp. (isolate 1 from Beninar, CZE), *Calothrix* sp. (isolate 1 from Pyhrabruck, AUT) and *Synechocystis* sp. (isolate 3 from IFA-Tulln, AUT) were grown in the 5-L tubular laboratory scale photobioreactor under photo-autotrophic non-sterile conditions in self-limiting 22O medium. In addition, IFA-3 and Heinrichs-4 were also cultivated outdoors in the same tubular photobioreactor without artificial illumination and without temperature control (see Appendix C, Figure A2). Examples of harvested dried biomasses are shown in Figure 6. *Synechocystis* sp. IFA-3 produced 7.9 g and *Chlorogloeopsis* sp. Heinrichs-4 produced 5.2 g dried biomass after three weeks cultivation in the 5-L photobioreactor containing 11.4 and 4.6% PHB, respectively.

The most noticeable outcomes of the upscaling experiments were the native resistance of *Synechocystis* sp. IFA-3 against grazing ciliates to which *Synechocystis* sp. CCALA192 and *Synechocystis* PCC6803, both were highly susceptible [19] and the wide cultivation temperature range between +4 and +45 °C of *Chlorogloeopsis* sp. Heinrichs-4. These two wild type isolates will be characterized more deeply and will be investigated for maximum growth and optimized PHB production. However, *Synechocystis* IFA-3 is a superior strain for outdoor cultivation in open low-tech photobioreactors in central Europe.

### 3.8. Obtaining Axenic Cultures

Different methods and combinations of methods were tested to obtain axenic cyanobacteria cultures. We started with chemical and growth-related methods, such as phototaxis, salt and temperature stress and applied different antibiotics techniques. All those did not yield us axenic cultures. Thus, we successively applied different physical methods, such as cell sorting, cell picking with a micromanipulator and a serial dilution method organized like the MPN method. Not one of those procedures did yield us axenic cultures on its own, only the sequential combination of all three physical methods succeeded.

#### 3.8.1. Phototaxis

Despite literature reports about robust positive phototaxis of *Synechocystis* sp. PCC 6803 [34], we did not observe any directed movement of cyanobacterial cells, neither on plates with 0.4% nor with 1% agar (Figure 7). In general, cyanobacteria and heterotrophic bacteria grew better on plates with 0.4% agar.

The procedure is described as a simple and effective method for phototaxis-exhibiting flagellates and cyanobacteria. In contrast, it is not applicable for species with similar swimming capabilities and it is not possible to eliminate attached bacteria [20].

#### 3.8.2. Antibiotic Treatment

In the first approach, carried out with *Synechocystis* PCC6803, neither cyanobacteria nor heterotrophic bacteria were reduced. In BG-I medium the cyanobacterial cell density increased from 1.2 × 10^7^ to 1.6 × 10^8^ mL^−1^, while heterotrophs (reference) increased from 1.6 × 10^7^ to 1.1 × 10^8^ mL^−1^. Even in BG-I+ medium cell densities increased—cyanobacteria from 1.2 × 10^7^ to 3.6 × 10^8^ mL^−1^, heterotrophic bacteria (reference) from 1.6 × 10^7^ to 3.3 × 10^8^ mL^−1^.

For the approach with the strains CCALA192 and *Synechocystis* sp. IFA-3 a similar picture was drawn. In BG-I CCALA192 increased from 4.0 × 10^6^ to 2.2 × 10^8^ cells mL^−1^ and heterotrophs from 3.5 × 10^6^ to 4.1 × 10^7^ cells mL^−1^. *Synechocystis* sp. IFA-3 and its accompanying flora increased from 3.0 × 10^6^ to 7.5 × 10^8^ cells mL^−1^ and from 3.0 × 10^6^ to 1.8 × 10^8^ cells mL^−1^, respectively. When glucose was added to the final medium the cell density of CCALA192 and heterotrophic bacteria increased as well—CCALA192: 1.0 × 10^6^ to 4.6 × 10^7^ cells mL^−1^, accompanying flora: 6.5 × 10^6^ to 1.7 × 10^7^ cells mL^−1^, the accompanying flora of *Synechocystis* sp. IFA-3: 2 × 10^6^ to 6.1 × 10^9^ cells mL^−1^. *Synechocystis* sp. IFA-3 in contrast did not grow after the antibiotic-treatment and the subsequent incubation in glucose containing medium.

#### 3.8.3. Elevated Temperature and Salinity

At 22 °C and in BG-I medium *Synechocystis* PCC6803 reached a cell density of 6.4 × 10^8^ cells mL^−1^ and the heterotrophic bacteria 4.2 × 10^8^ cells mL^−1^. At 37 °C *Synechocystis* PCC6803 reached 6.4 × 10^8^ cells mL^−1^, but heterotrophs just 1.2 × 10^8^ cells mL^−1^. In BG-I+ medium (with glucose), heterotrophic bacteria had an advantage at 37 °C and reached 2.5 × 10^9^ cells mL^−1^ instead of 1.5 × 10^9^ cells mL^−1^ at 22 °C. It can be said that *Synechocystis* PCC6803 did not mind higher temperatures (37 °C), but the isolated cyanobacterial accompanying flora did (Figure 8).

*Synechocystis* sp. IFA-3 reached a cell density of 1.3 × 10^8^ mL^−1^ in BG-I medium without salt at 22 °C, whereas with 4% NaCl just 1.6 × 10^7^ cells mL^−1^ were reached. At 37 °C 6.6 × 10^8^ and 5.0 × 10^6^ cells mL^−1^ were obtained in 0% and 4% NaCl, respectively. The starting cell density was 1.2 × 10^7^ cells mL^−1^. A similar picture could be drawn for the heterotrophic cells which started at a cell density of 9 × 10^6^–1 × 10^7^ mL^−1^. In the media with salt heterotrophic, cells did not grow at all, in BG-I at 22 °C they grew best and reached a cell density of 5.8 × 10^7^ mL^−1^, at 37 °C they just reached 1.1 × 10^7^ mL^−1^. These results were also visible with bare eyes (Figure 9).

Methods using antibiotics or lysozyme are so called killing methods. According to Vu et al. (2018) advantages of these methods are that they selectively kill contaminations over algae cells and that they can be combined with other methods to increase efficiency. Lysozyme can be used as an alternative for antibiotic-sensitive algae. Antibiotics can only be used to a limited extent for cyanobacteria, compounds are often toxic/mutagenic to algae. Additionally, beforehand the appropriate antibiotic type, concentration and treating time need to be evaluated and there is concern about the environmental impact of the spread of antibiotic-resistant microbes. In case of lysozyme, long exposure time needs to be considered since the cell membrane can be weakened [20].

#### 3.8.4. Cell Sorter

Cultures of the three *Synechocystis* strains (PCC6803, CCALA192 and IFA-3) were counted and separated by cell size and chlorophyll fluorescence intensity into two collected and one waste fraction. The waste fraction was almost colorless and contained mostly heterotrophic bacteria and unidentified cell fragments (controlled by microscopy) while cyanobacteria of the typical spherical cell shape, but of different sizes (3–4 and 5–6 µm) were found in the other two fractions. The cyanobacteria fractions contained roughly one million cells each.

After cultivation in BG-I medium for two weeks, the cultures were inspected and turned out to be still not axenic. Furthermore, cell sizes were distributed in the same ranges as before, independent from the cultivated fraction. A second cell sorting with each of those cultures gave a similar, almost identical separation into one waste and two fractions of *Synechocystis* sp. in two cell size classes.

While cell sorting appeared to be not suitable for obtaining axenic cultures, the meaning of parallel existence of two cell sizes in a culture of a single-species *Synechocystis* strain remains unclear.

#### 3.8.5. Micromanipulator

Several attempts were made to pick a small number of cyanobacteria cells from a single-species non axenic culture of each of the three *Synechocysitis* strains (PCC6803, CCALA 192 and IFA-3). The number of picked cells was always in the range of 5 to 20.

As described by Vu et al., 2018 [20], it was hard to catch the tiny *Synechocystis* cells but avoid catching heterotrophic bacteria. Therefore, only the cultures obtained from the commercial culture collections and those which had comparably low numbers of heterotrophic bacteria from the selection cultures (as described in Section 3.2, Section 3.3 and Section 3.4) were used.

We achieved new cultures of all three strains that contained a lower variety of heterotrophic bacteria (compared by cell sizes and shapes from microscopic inspection), but we were not able to obtain axenic cultures with this method.

#### 3.8.6. MPN Dilution Approach

Serial dilutions have the advantage of being simple and suitable to isolate new algal species from environmental samples. According to Vu et al. (2018), only monoculture levels can be achieved and this method is only suitable for organisms that are abundant in the sample and is largely ineffective for rare organisms [20].

We used the previously enriched cyanobacteria cultures of the three *Synechocystis* strains PCC6803, CCALA192 and IFA-3, which we obtained after the micromanipulator process. This increased the chance to get single cyanobacteria cells separated from the, meanwhile, lower number of heterotrophic bacteria distributed over the 24 wells of the plate. Cultivation times were four to six weeks for each trial until enough cells were grown in the wells to allow microscopic control. Putative axenic cultures were transfer into shaking flasks with BG-I medium and were controlled weekly.

After several separation cycles we obtained axenic cultures from PCC6803 and CCALA192 and one culture from *Synechocystis* sp. IFA-3 which contained a low number of only one coccoid heterotrophic bacteria type of ca. 1–1.5 µm diameter. We have strong evidence that these bacteria are obligate synergists to *Synechocystis* sp. IFA-3, as all further trials to separate them resulted either in a culture with the heterotrophic bacteria present or in no growth at all.

### 3.9. All Efforts from Isolation to Axenic Cultures in Summary

The choice of sampling sites is always determined by the goal to be achieved. In our case, we hoped to find strains which were adapted to warm (Mediterranean) and other adapted to cool or cold (alpine) climate.

A selection for fastest growing strains by frequent transfer of mixed cultures into new mineral medium was the most obvious procedure to search for potential production strains suitable for biotechnological processes and was comparably easy to achieve. The frequent transfers reduced heterotrophs, as accumulation of dead cells (and of potential organic substrate) was held at a minimum.

Getting rid of eukaryotic microorganisms succeeded in the majority of cases at the first attempt by cycloheximide addition. However, we lost about one third of the isolates which previously contained fast growing cyanobacteria due to the application of antibiotics. Especially all of the samples deriving from habitats in cold environments showed increased susceptibility.

The highest effort of all was the purification to axenic cultures. Especially all of the chemical (media composition, salinity), the temperature and the phototaxis approaches failed completely. By empirical trial and error attempts, we were able to reduce the amount and diversity (checked by morphological criteria via light microscopy) of the accompanying flora. The steps were a twofold sequential separation of cells by size and optical properties in the cell sorter, followed by multiple sequential cell picks with the micromanipulator. This resulted in cultures that showed a significantly reduced number of cell shapes in comparably low abundances as remaining accompanying flora and which were dominated by the cyanobacteria cells. The dilution method was the final necessary step to obtain axenic cultures from those pre-purified cultures, and even this method had a high failure rate.

We succeeded in getting axenic *Synechocystis* cultures from the non-axenic products obtained from commercial culture collection strains PCC6803 and CCALA192. The wild type IFA-3 refused to grow as axenic culture, its accompanying coccoidal bacteria are, most probably, obligate synergists.

## 4. Conclusions

Sampling 25 locations from so different sites as a geyser in Iceland, a wet rock in the Dolomite Alps, a dammed lake in southern Spain, a glacier lake in Argentina and a snow field in Greenland resulted in 71 primary cultures of which 34 were growing stable over multiple medium transfers. 19 among those were purified into single-species cyanobacteria strains and identified. Only four of them were producing and accumulating PHA or PHB and were tested in a tubular laboratory photobioreactor, and only two are promising production strains that can be cultivated in open non-axenic photobioreactors without artificial illumination and without temperature control.

The lessons learned from three years of collecting, cultivating and characterizing wild type cyanobacteria strains and two additional years of preparing axenic cultures are (1) to be patient and (2) to purify only those isolates into axenic cultures that are intended for advanced molecular analysis, such as transcriptomics or genetic manipulations. We achieved production of biomass and PHA or PHB under rough outdoor conditions from non-axenic strains with acceptable biomass growth and with promising PHA or PHB production rates. For the purpose of this work, it was of primary importance to find strains that are adapted to the local climate and can be grown outdoors with the lowest effort for process control. Sure, we would have been happy to find a strain that would have outgrown all others while accumulating the highest PHA or PHB amounts. However, that did not happen.

It is not without a certain irony that among the samples collected from almost all over the world, a strain isolated from the fire pond in front of the institute building showed the most rapid growth and the best PHB production characteristics.

Axenic cultures and *Synechocystis* sp. IFA-3 will be deposited at Culture collection of Autotrophic Organisms (CCALA). Investigations on the putative obligate synergism of the IFA-3 culture is already in work, and the upscaling and growth optimization of the *Chlorogloeopsis* Heinrichs-4 isolate will be the topic of upcoming research.

## Figures and Tables

**Figure 1 bioengineering-09-00178-f001:**
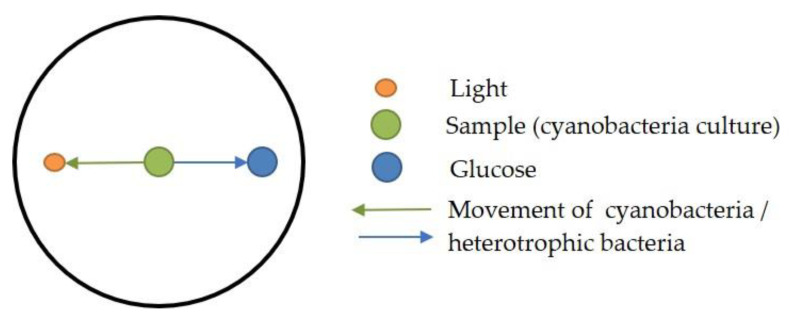
Agar-plate with the locations for sample (green) and glucose (blue) as well as the spot where light reaches the plate (orange) as well as the expected movement of cyanobacteria and heterotrophic bacteria (arrows).

**Figure 2 bioengineering-09-00178-f002:**
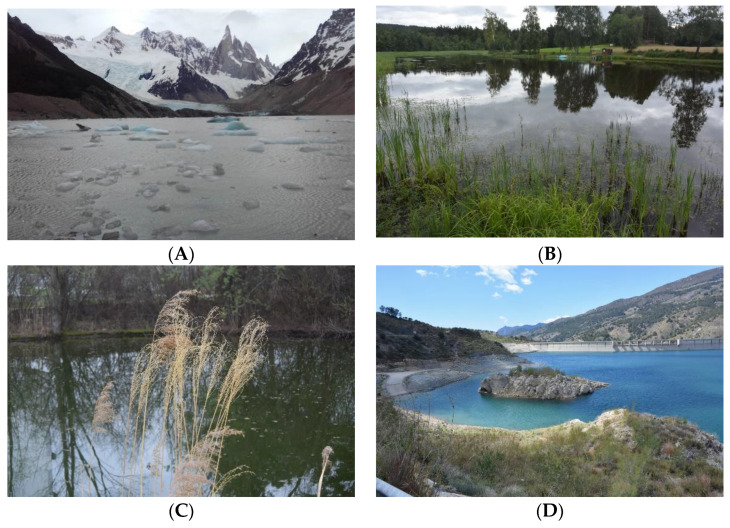
Example sampling locations of different climates: (**A**) Los Glaciares Laguna Torre ARG; (**B**) Heinrichs AUT (isolate: *Chlorogloeopsis fritschii*; (**C**) Tulln IFA-Teich AUT (isolate: *Synechocystis* sp.); and (**D**) Presa de Beninar ESP (isolate: *Crocosphaera* sp.).

**Figure 3 bioengineering-09-00178-f003:**
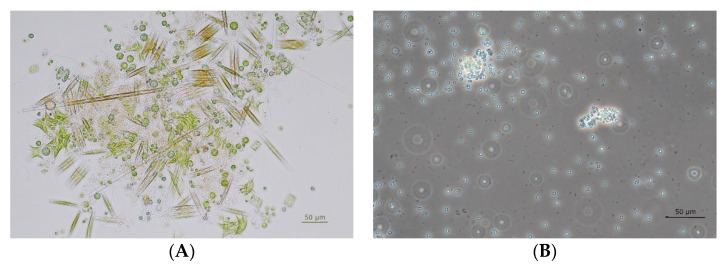
Microscopic control of phtoto-autotrophically cultivated samples from the fire-pond at IFA-Tulln (AUT); (**A**) original mixed culture enriched in BG-I mineral medium (bright field image) and (**B**) after cultivation in presence of cycloheximide (phase contrast image).

**Figure 4 bioengineering-09-00178-f004:**
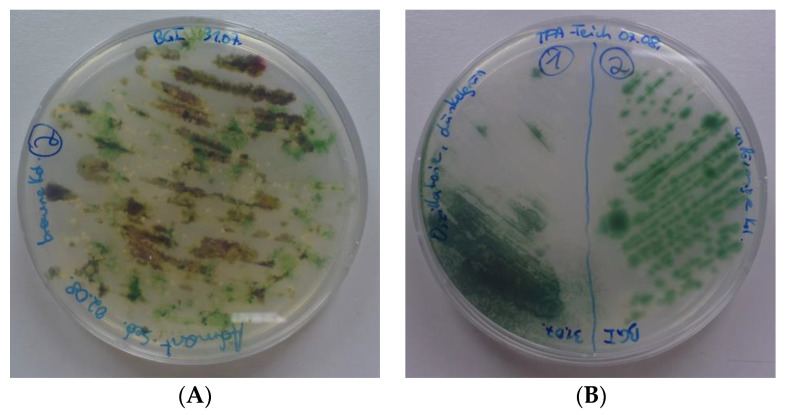
Fractionated streaks of mixed cyanobacteria samples after cycloheximide treatment taken from (**A**) Admont (AUT) and (**B**) Tulln, IFA-Teich (AUT).

**Figure 5 bioengineering-09-00178-f005:**
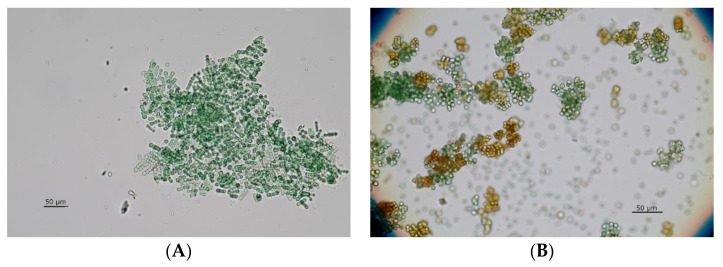
Microscopic imaging of *Chlorogloeopsis* sp. (Heinrichs-4, AUT) cultures; (**A**) blue-green cells during the growth phase and (**B**) in the stage of nitrogen and phosphorus limitation; (**C**) aged culture in stationary phase in bright field visible light and (**D**) the same section in fluorescence mode after staining with Nile-red.

**Figure 6 bioengineering-09-00178-f006:**
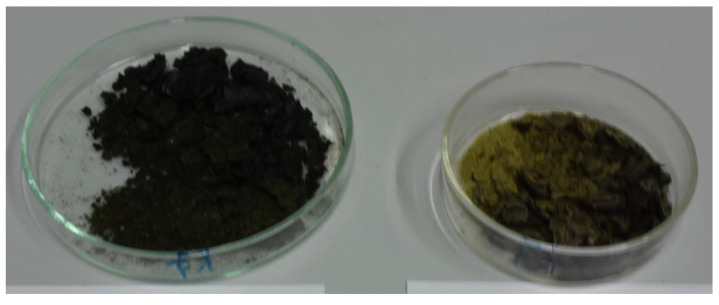
Dried biomasses from cultivation of *Synechocystis* sp. IFA-3 (**left**) and *Chlorogloeopsis* sp. Heinrichs-4 (**right**) in the 5-L tubular photobioreactor.

**Figure 7 bioengineering-09-00178-f007:**
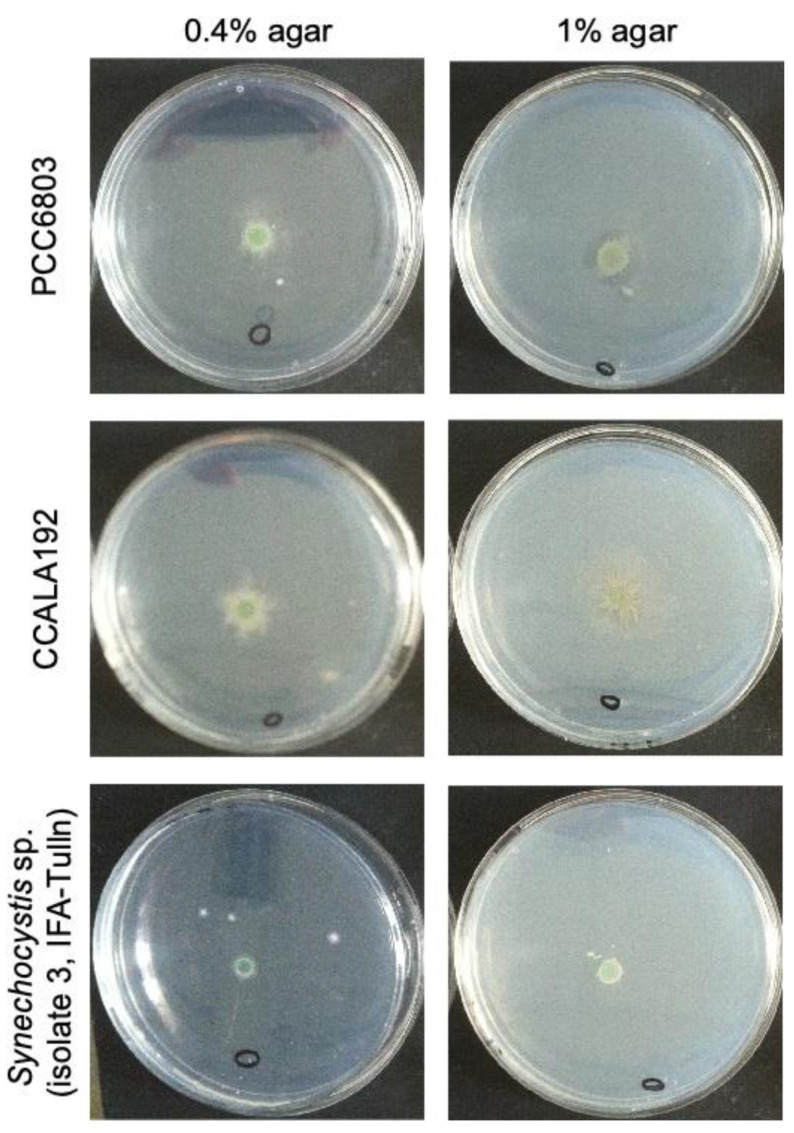
Results of the phototaxis experiment with cultures of PCC6803 (**top**), CCALA192 (**middle**) and *Synechocystis* sp. IFA-3 (**bottom**) on agar plates with 0.4% (**left**) and 1% agar (**right**). Circle: spot where glucose was applied.

**Figure 8 bioengineering-09-00178-f008:**
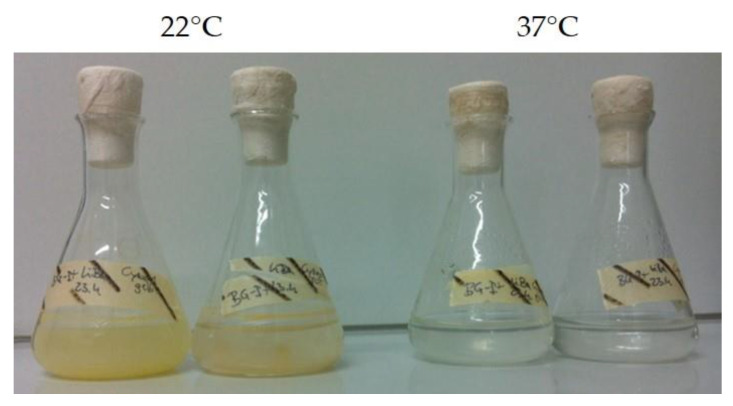
Growth of isolated cyanobacterial accompanying flora inoculated in BG-I medium at room temperature (22 °C) and elevated temperature (37 °C).

**Figure 9 bioengineering-09-00178-f009:**
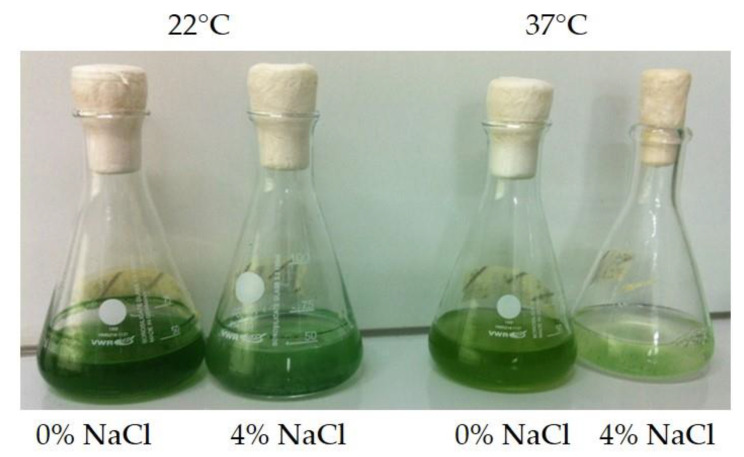
*Synechocystis* sp. IFA-3 inoculated in BG-I with and without NaCl at room temperature (22 °C) and elevated temperature (37 °C).

**Table 1 bioengineering-09-00178-t001:** Sampling locations with climate types and identified single-species strains. PHB production was tested by cultivation in a laboratory photobioreactor of ca. 5 L volume. Sequences are provided in Appendix B.

Sampling Location	Coordinates	Climate Region ^1^	No. of Samples	Identification via 16S rDNA	Dry Biomass and PHB Content
Admont (AUT)	47°34′51.0″ N 14°27′13.5″ E	Dfc	4	*Pseudanabaena* sp. [99.6%]	n.d.
Beagle Channel (ARG)	54°50′59.7″ S 68°29′37.9″ W	ET	1	n.d.	n.d.
Branná (CZE)	50°09′24.6″ N 17°01′17.1″ E	Dfb	7	*Calothrix* sp. [99%], *Pseudanabaena* [97%]	n.d.
Chlum & Třeboně (CZE)	48°57′33.1″ N 14°56′03.3″ E	Cfb	6	n.d.	n.d.
Elisabethsee Amerbach (AUT)	47°10′43.4″ N 12°31′44.7″ E	ET	1	n.d.	n.d.
Glaciar Perito Moreno (ARG)	50°29′19.6″ S 73°03′35.9″ W	Cfc/ET	1	n.d.	n.d.
Gmünd, Mondteich (AUT)	48°46′40.3″ N 14°59′51.5″ E	Cfb	2	*Calothrix desertica* [97%]	n.d.
Greenland (GRL)	n.d.	EF/ET	5	*Cyanobium gracile* [99.8%]	n.d.
Greifenstein, Danube (AUT)	48°20′44.7″ N 16°14′19.3″ E	Cfb	2	*Pseudoanabaena biceps* [98.7%]	n.d.
Heidenreichstein, Hofwehrteich, (AUT)	48°52′06.1″ N 15°07′44.2″ E	Dfb/Cfb	1	n.d.	n.d.
Heinrichs (AUT)	48°44′56.3″ N 14°50′07.9″ E	Dfb/Cfb	5	*Calothrix* sp. [98%]	n.d.
				*Chlorogloeopsis fritschii* [99.8%]	1040 mg L^−1^, 4.6% PHB
Island (ISL)	64°18′45.2″ N 20°18′08.3″ W	Cfc/ET	1	n.d.	n.d.
Island (ISL)	63°54′53.8″ N 22°41′50.9″ W	Cfc/ET	1	n.d.	n.d.
Lago di San Vito (ITA)	46°28′03.8″ N 12°12′06.4″ E	Dfc/ET	1	n.d.	n.d.
Langenrohr, gr. Tulln (AUT)	48°19′06.8″ N 16°01′00.4″ E	Cfb	3	*Pseudoanabaena biceps* [99.7%]	n.d.
				*Cyanobium gracile* [99.6%]	n.d.
National park los glaciares, Laguna Torre (ARG)	49°19′47.4″ S 72°59′23.4″ W	Cfc/ET	1	n.d.	n.d.
National park los glaciares, Lagunas Madre e hija (ARG)	49°18′18.8″ S 72°57′03.7″ W	Cfc/ET	1	n.d.	n.d.
Passo di Falzarego (ITA)	46°31′09.1″ N 12°00′32.0″ E	Dfc/ET	1	n.d.	n.d.
Passo della Guardia (ITA)	44°03′04.7″ N 7°44′41.5″ E	Csb	1	n.d.	n.d.
Presa de Beninar (ESP)	36°52′42.3″ N 3°01′31.6″ W	Csa	3	*Crocosphaera* sp. [96%]	<1% PHB
				*Synechococcus* sp. [96%]	n.d.
Pyhrabruck (AUT)	48°46′11.0″ N 14°49′16.3″ E	Cfb	3	*Calothrix* sp. [97%]	16.8 mg L^−1^ 1.1% PHB
Río Cacheuta (ARG)	33°02′53.5″ S 69°11′47.3″ W	BSk/BWk	1	n.d.	n.d.
Río De la Cascada (ARG)	49°17′37.0″ S 72°54′08.6″ W	Cfc/ET	1	n.d.	n.d.
Río Mendoza (ARG)	32°56′13.5″ S 69°12′25.6″ W	BSk/BWk	1	n.d.	n.d.
Thermal SPA Loipersdorf (AUT)	46°59′11.2″ N 16°06′36.9″ E	Cfb	12	*Anabaena Trichormus variabilis* [98.8%]	n.d.
				*Nostoc* sp. [98.5%]	n.d.
				*Nodosilinea nodulosa* [97.9%]	n.d.
				*Anabaena variabilis* [98.8%]	n.d.
Tulln, IFA-Teich (AUT)	48°19′14.5″ N 16°03′59.2″ E	Cfb	5	*Calothrix* sp. [100%]	n.d.
				*Synechocystis* sp. [98.2%]	1680 mg L^−1^, 11.4% PHB
				*Pseudanabaena biceps* [99.3%]	n.d.

^1^ According to Köppen climate classification [33]: BSk: Cold semi-arid climate, BWk: Cold desert climate, Cfb: Temperate oceanic climate, Cfc: Subpolar oceanic climate, Csa: Hot-summer Mediterranean climate, Csb: Warm-summer Mediterranean climate, Dfb: Warm-summer humid continental climate, Dfc: Subarctic climate, EF: Ice cap climate, ET: Tundra climate, n.d.: no data available.

## Data Availability

Not applicable.

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
