# Peer review of "PHB Producing Cyanobacteria Found in the Neighborhood—Their Isolation, Purification and Performance Testing"

_bioengineering, 2022, doi:10.3390/bioengineering9040178_

Round 1

Reviewer 1 Report

The article entitled  PHB producing cyanobacteria found in the neighborhood - their isolation, purification and performance testing  submitted to Bioengineering Journal. This manuscript is generally well written and clearly presented however still needs to address some comments, and thus require substantial major revision to improve the quality of the manuscript.

1)      Provide a nice graphical abstract representing the overview of the MS with key highlights. 

2)     Abstract looks very general and not informative. In abstract authors should mention is this PHA or PHB? Add importance of research work in one or two sentences. 

Without knowing the characteristics how author can say that the produced PHA is PHB?

3)      In the introduction section, write the novelty of the work and the problem statement clearly. Authors fails to explain the plastic pollution and their quantitative details for this author can refer to the recent review article Bioresource technology 325, 124685, 2021.

Both Figure 1 are not necessary.

4)      Write the practical applications and future research perspectives and challenges by adding a new section before conclusions

5)      The conclusion of the study is not discussed with the specific output obtained from the study, it could be modified with precise outcomes with a take home message.

6)      English and grammar mistakes are present. The author should check the manuscript by native English Speaker to improve the quality of the manuscript.

Author Response

[Reviewer 1 comment in black / authors response in blue]

Comments and Suggestions for Authors

The article entitled PHB producing cyanobacteria found in the neighborhood - their isolation, purification and performance testing submitted to Bioengineering Journal. This manuscript is generally well written and clearly presented however still needs to address some comments, and thus require substantial major revision to improve the quality of the manuscript.

We thank Reviewer 1 for the constructive comments that improved our manuscript to a great extent. We revised our manuscript in following the recommendations as described below.

1) Provide a nice graphical abstract representing the overview of the MS with key highlights.

The graphical abstract was originally submitted together with the manuscript at page 2 (which was wrong). Additionally, we uploaded it as separate file (which had been the right way to do). We have, again, uploaded the graphical abstract as a separate file together with the revised manuscript, not including it, not numbering it.

2) Abstract looks very general and not informative. In abstract authors should mention is this PHA or PHB? Add importance of research work in one or two sentences.

We re-formulated parts of the abstract accordingly. Furthermore, a statement (with reference) about cyanobacteria producing the homo-polymer PHB is included in the Introduction (see below).

Without knowing the characteristics how author can say that the produced PHA is PHB?

We added two references (one recently published, one from 1982) which, by themselves, refer to many other publications, old an new, clearly stating that the homo-polymer PHB is with >99% the dominant PHA synthesized by all ever investigated cyanobacteria when grown photo-autotrophically. In addition we refer to our own findings (Kovalcik et al, 2017) were we analysed PHB from Synechocystis CCALA192. Indeed, we didn‘t analyse the PHA composition ofChlorogloeopsis sp. (Heinrichs-4) and, therefore, changed the description accordingly.

3) In the introduction section, write the novelty of the work and the problem statement clearly. Authors fails to explain the plastic pollution and their quantitative details for this author can refer to the recent review article Bioresource technology 325, 124685, 2021.

We reformulated parts of the Introduction section and added one paragraph to make the intended goal of our work more clear. After serious consideration and discussion, we are willing to include the requested reference of Biores. Tech. 325, 124685, although our work does NOT intend to utilise waste streams from biofuel, biodiesel or whey. Our work does focus on the isolation of locally adapted phototrophic cyanobacteria and on the characterisation of those microorganisms for their use as PHB producers. We hope very much, Reviewer 1 is not among the authors of the requested reference.

Both Figure 1 are not necessary.

Unfortunately, there was some irregularity and mismatch with automatic numbering of graphs and tables, which happened during upload and was not recognized immediately. The fields were deleted and replaced with manual numbering. We apologize for the confusion! We deleted the photo of the micromanipulator but kept the drawing of the phototaxis principle, because two other Reviewers liked it (following a majority vote). As mentioned before, we removed the graphical abstract (by error also named figure 1) from the main document and uploaded it separately (as requested).

4) Write the practical applications and future research perspectives and challenges by adding a new section before conclusions

We added more text (section 3.9) to explain what we learned from our experiments and how we, meanwhile, have transferred our knowledge into new research and applications. However, we split this information between discussion and conclusion.

5) The conclusion of the study is not discussed with the specific output obtained from the study, it could be modified with precise outcomes with a take home message.

We re-formulated the conclusion section and put more emphasis on the output of our work and on its meaning for research and developments in PHA/PHB production with phototrophic microorganisms.

6) English and grammar mistakes are present. The author should check the manuscript by native English Speaker to improve the quality of the manuscript.

We proof-read the revised manuscript and, indeed, found and eliminated multiple typos and grammar mistakes.

Reviewer 2 Report

The paper contains some interesting results and provided advancement in scientific knowledge.

Use of more recent references in the introduction is strongly recommended, For example: Journal of Environmental Chemical Engineering 9 (2021) 105379; Heliyon 6 (2020) e05381

Science of the Total Environment 800 (2021) 149561; Processes 8 (2020) 323; Bioresource Technology 289 (2019) 121700

Author Response

[Reviewer 2 comment in black / authors response in blue]

Comments and Suggestions for Authors

The paper contains some interesting results and provided advancement in scientific knowledge.

We thank Reviewer 2 and we are pleased by the friendly and motivating comment.

Use of more recent references in the introduction is strongly recommended, For example: Journal of Environmental Chemical Engineering 9 (2021) 105379; Heliyon 6 (2020) e05381

Science of the Total Environment 800 (2021) 149561; Processes 8 (2020) 323; Bioresource Technology 289 (2019) 121700

Indeed, we used not enough references to position our work. We added the suggested (and some more) references to our revised manuscript and especially refer to the cyanobacteria specific publications. Further, we added our impression and our thoughts about the many cyanobacteria species which were identified and characterized in the past as PHB producers. However, none of those had been tested in open (non-sterile) vessels in bigger volume (at ambient outdoors conditions). We had repeated contact to some of the referred authors before and discussed with them the specific difficulties in upscaling. And we all agreed upon that cyanobacteria cultivated in sterile shaking flasks can find highly promising producers which fail at outdoor conditions in non-sterile photobioreactors. This is practical experience which is hard to tell in a manuscript and impossible to find references (who likes to publish failures?) but is, nevertheless, a serious observation with consequences for technology transfer.

We inserted a short statement in the introduction and also added a new section 3.9 and two paragraphs in the conclusion.

Reviewer 3 Report

In this manuscript, the authors mainly conducted isolation of cyanobacterial strains from various environments, and successfully established 19 strains. Further, the authors confirmed some of them could produce PHB under the nutrient-limited conditions. As the second topic, the authors tested several methods (cellsorter, phototaxis, antibiotics, salinity stress, micromanipulation, and MPN) to establish axenic culture strains; several approaches do not seem to work for Synechocystis strains (PCC6803, CCALA192 and IFA-3). I think the manuscript is well written as a communication paper. I list my comments below.

<major comments>
1) The authors applied six approaches to obtain axenic cultures of Synechocystis. The results for each approach are described in 3.8.1 to 3.8.6, but it is unclear what the suitable methods are. I’d like to suggest to summarize all results (possibly in Table) and compare them to clarify the best method to establish axenic cultures of cyanobacteria. 

2) All sequenced data of 16S rDNA should be accessible. Please deposit them in a database (such as GenBank) and indicate the accession numbers with the strain name in Table 1.

<Minor changes>
“sp.” should not be italicized.
Line 44. “immission” will be “emission”
Line 77. “heterotrophe” will be “heterotroph”
Line 123. “1,5%” will be “1.5%”.
Line 270. “Figure 1” will be “Figure 2”.
Line 295. “in Table 2” will be “in Table 1”.
Line 309. “Figure 2” will be “Figure 4”.
Line 314. “Algae free cultures were obtained in most cases”. Cyanobacteria are also algae. 
Line 334. “Figure 3” will be “Figure 5”.
Line 340. “Table 2” will be “Table 1”.
Line 349. “granula” will be “granules”.
Line 352. “Table 2” will be “Table 1”.
Line 353. “Figure 4” will be “Figure 6”.
Line 371. “Figure 5” will be “Figure 7”.
Line 419. “Figure 6” will be “Figure 8”. The pictures are too small, and synechocystis colonies are not visible. 
Line 452. “Figure 7” will be “Figure 9”.
Line 459. “cells did not grew” will be “cells did not grow”.
Line 471. “Figure 8” will be “Figure 10”.
Line 490. “32.” Will be “3.2”.

Author Response

[Reviewer 3 comment in black / authors response in blue]

Comments and Suggestions for Authors

In this manuscript, the authors mainly conducted isolation of cyanobacterial strains from various environments, and successfully established 19 strains. Further, the authors confirmed some of them could produce PHB under the nutrient-limited conditions. As the second topic, the authors tested several methods (cellsorter, phototaxis, antibiotics, salinity stress, micromanipulation, and MPN) to establish axenic culture strains; several approaches do not seem to work for Synechocystis strains (PCC6803, CCALA192 and IFA-3). I think the manuscript is well written as a communication paper. I list my comments below.

We are pleased by the detailed comments of Reviewer 3 and are willing to follow the suggested changes which will improve the manuscript by a great extent.

<major comments>

1) The authors applied six approaches to obtain axenic cultures of Synechocystis. The results for each approach are described in 3.8.1 to 3.8.6, but it is unclear what the suitable methods are. I’d like to suggest to summarize all results (possibly in Table) and compare them to clarify the best method to establish axenic cultures of cyanobacteria.

Indeed, while the gone way was clear to us, the explanation was not that clearly formulated. We added a small paragraph at the beginning and another at the end of the section dealing with axenic cultures. Intermediately we tried to make a table as suggested, but then failed in making it readable while still including essential information. We hope, Reviewer 3 can accept our attempt with a literal description.

2) All sequenced data of 16S rDNA should be accessible. Please deposit them in a database (such as GenBank) and indicate the accession numbers with the strain name in Table 1.

The life cultures of Synechocystis CCALA192 (axenic) and Synechocystis IFA-3 (non axenic) will be deposited at the CCALA culture collection and will be allowed for public use.

We deposited the data of only two strains in a database but did not with all the others. For a simple reason: we did not keep the non-suitable strains (those which did produce no or not sufficient amounts of PHB). Those can‘t be accessed any longer. For documentation we added a third annex to this manuscript where we list the 16S-rDNA sequences of all cyanobacteria isolates.

<Minor changes>

“sp.” should not be italicized.

Thank you! We overlooked that!

Line 44. “immission” will be “emission”

In this case immission was meant correctly. The amount of incoming light is the essential factor.

Line 77. “heterotrophe” will be “heterotroph”

Line 123. “1,5%” will be “1.5%”.

Line 270. “Figure 1” will be “Figure 2”.

Line 295. “in Table 2” will be “in Table 1”.

Line 309. “Figure 2” will be “Figure 4”.

Line 314. “Algae free cultures were obtained in most cases”. Cyanobacteria are also algae.

Line 334. “Figure 3” will be “Figure 5”.

Line 340. “Table 2” will be “Table 1”.

Line 349. “granula” will be “granules”.

Line 352. “Table 2” will be “Table 1”.

Line 353. “Figure 4” will be “Figure 6”.

Line 371. “Figure 5” will be “Figure 7”.

Line 419. “Figure 6” will be “Figure 8”. The pictures are too small, and synechocystis colonies are not visible.

Line 452. “Figure 7” will be “Figure 9”.

Line 459. “cells did not grew” will be “cells did not grow”.

Line 471. “Figure 8” will be “Figure 10”.

Line 490. “32.” Will be “3.2”.

We happily followed all of those suggested corrections and express our thanks to Reviewer 3 for recognizing and reporting the errors. Unfortunately, there was some irregularity and mismatch with automatic numbering of graphs and tables, which happened during upload and which we did not recognize imediately. The MS-word fields were deleted in the revised version and replaced with manual numbering. We apologize for the confusion and for the increased effort for the review!

Round 2

Reviewer 1 Report

The authors have substantially revised the manuscript according to the comments.

The present form of the manuscript can be accepted for publication.

Congratulations !!